# Is a Good Representation Sufficient for Sample Efficient Reinforcement Learning?

**Simon S. Du**
Institute for Advanced Study
ssdu@ias.edu

**Sham M. Kakade**
University of Washington, Seattle
sham@cs.washington.edu

**Ruosong Wang**
Carnegie Mellon University
ruosongw@andrew.cmu.edu

**Lin F. Yang**
University of California, Los Angles
linyang@ee.ucla.edu

## ABSTRACT

Modern deep learning methods provide effective means to learn good representations. However, is a good representation itself sufficient for sample efficient reinforcement learning? This question has largely been studied only with respect to (worst-case) approximation error, in the more classical approximate dynamic programming literature. With regards to the statistical viewpoint, this question is largely unexplored, and the extant body of literature mainly focuses on conditions which *permit* sample efficient reinforcement learning with little understanding of what are *necessary* conditions for efficient reinforcement learning.

This work shows that, from the statistical viewpoint, the situation is far subtler than suggested by the more traditional approximation viewpoint, where the requirements on the representation that suffice for sample efficient RL are even more stringent. Our main results provide sharp thresholds for reinforcement learning methods, showing that there are hard limitations on what constitutes good function approximation (in terms of the dimensionality of the representation), where we focus on natural representational conditions relevant to value-based, model-based, and policy-based learning. These lower bounds highlight that having a good (value-based, model-based, or policy-based) representation in and of itself is insufficient for efficient reinforcement learning, unless the quality of this approximation passes certain hard thresholds. Furthermore, our lower bounds also imply exponential separations on the sample complexity between 1) value-based learning with perfect representation and value-based learning with a good-but-not-perfect representation, 2) value-based learning and policy-based learning, 3) policy-based learning and supervised learning and 4) reinforcement learning and imitation learning.

## 1 INTRODUCTION

Modern reinforcement learning (RL) problems are often challenging due to the huge state space. To tackle this challenge, function approximation schemes are often employed to provide a compact representation, so that reinforcement learning can generalize across states. A common paradigm is to first use a feature extractor to transform the raw input to features (a succinct representation) and then apply a linear predictor on top of the features. Traditionally, the feature extractor is often handcrafted (Sutton & Barto, 2018), while more modern methods often train a deep neural network to extract features. The hope of this paradigm is that, if there exists a good low dimensional (linear) representation, then efficient reinforcement learning is possible.

Empirically, combining various RL function approximation algorithms with neural networks for feature extraction has lead to tremendous successes on various tasks (Mnih et al., 2015; Schulman et al., 2015; 2017). A major problem, however, is that these methods often require a large amount of samples to learn a good policy. For example, deep $Q$-network requires millions of samples to solve certain Atari games (Mnih et al., 2015). Here, one may wonder if there are fundamental statistical

limitations on such methods, and, if so, under what conditions it would be possible to efficiently learn a good policy?

In the supervised learning context, it is well-known that empirical risk minimization is a statistically efficient method when using a low-complexity hypothesis space (Shalev-Shwartz & Ben-David, 2014), e.g. a hypothesis space with bounded VC dimension. For example, polynomial number of samples suffice for learning a near-optimal $d$-dimensional linear classifier, even in the agnostic setting[1]. In contrast, in the more challenging RL setting, we seek to understand if efficient learning is possible (say from a sample complexity perspective) when we have access to an accurate (and compact) parametric representation — e.g. our policy class contains a near-optimal policy or our hypothesis class accurately approximates the optimal value function. In particular, this work focuses on the following question:

**Is a good representation sufficient for sample-efficient reinforcement learning?**

This question has largely been studied only with respect to approximation error in the more classical approximate dynamic programming literature, where it is known that algorithms are stable to certain worst-case approximation errors. With regards to sample efficiency, this question is largely unexplored, where the extant body of literature mainly focuses on conditions which are *sufficient* for efficient reinforcement learning though there is little understanding of what are *necessary* conditions for efficient reinforcement learning. In reinforcement learning, there is no direct analogue of empirical risk minimization as in the supervised learning context, and it is not evident what are the statistical limits of learning based on properties of our underlying hypothesis class (which may be value-based, policy-based, or model-based).

Many recent works have provided polynomial upper bounds under various sufficient conditions, and in what follows we list a few examples. For value-based learning, the work of Wen & Van Roy (2013) showed that for *deterministic systems*[2], if the optimal $Q$-function can be *perfectly* predicted by linear functions of the given features, then the agent can learn the optimal policy exactly with polynomial number of samples. Recent work (Jiang et al., 2017) further showed that if certain complexity measure called *Bellman rank* is bounded, then the agent can learn a near-optimal policy efficiently. For policy-based learning, Agarwal et al. (2019) gave polynomial upper bounds which depend on a parameter that measures the difference between the initial distribution and the distribution induced by the optimal policy.

**Our Contributions.** This paper gives, perhaps surprisingly, strong *negative* results to this question. The main results are *exponential lower bounds* in terms of planning horizon $H$ for value-based, model-based, and policy-based algorithms with given good representations[3]. Notably, the requirements on the representation that suffice for sample efficient RL are even more stringent than the more traditional approximation viewpoint. A comprehensive summary of previous upper bounds and our lower bounds is given in Table 1, and here we briefly summarize our hardness results.

1. For value-based learning, we show even if $Q$-functions of all policies can be approximated by linear functions of the given representation with approximation error $\delta = \Omega\left(\sqrt{\frac{H}{d}}\right)$ where $d$ is the dimension of the representation and $H$ is the planning horizon, then the agent still needs to sample exponential number of trajectories to find a near-optimal policy.

2. For model-based learning, we show even if the transition matrix and the reward function can be approximated by linear functions of the given representation with approximation error $\delta = \Omega\left(\sqrt{\frac{H}{d}}\right)$ (in $\ell_\infty$ sense), the agent still needs to sample exponential number of trajectories to find a near-optimal policy.

3. We show even if optimal policy can be *perfectly* predicted by a linear function of the given representation with a strictly positive margin, the agent still requires exponential number of trajectories to find a near-optimal policy.

---

[1]Here we only study the sample complexity and ignore the computational complexity.

[2]MDPs where both reward and transition are deterministic.

[3]Our results can be easily extend to infinite horizon MDPs with discount factors by replacing the planning horizon $H$ with $\frac{1}{1-\gamma}$, where $\gamma$ is the discount factor. We omit the discussion on discount MDPs for simplicity.

These lower bounds hold even in deterministic systems and even if the agent knows the transition model. Note these negative results apply to the case where the $Q$-function, the model, or the optimal policy can be predicted well by a linear function of the given representation. Since the class of linear functions is a strict subset of many more complicated function classes, including neural networks in particular, our negative results imply lower bounds for these more complex function classes as well. Our results highlight the following conceptual insights:

- The requirements on the representation that suffice for sample efficient RL are significantly more stringent than the more traditional approximation viewpoint; our statistical lower bounds show that there are hard thresholds on the worst-case approximation quality of the representation which are not necessary from the approximation viewpoint.

- Since our lower bounds apply even when the agent knows the transition model, the hardness is not due to the difficulty of exploration in the standard sense. The unknown reward function is sufficient to make the problem exponentially difficult.

- Our lower bounds are not due to the agent's inability to perform efficient supervised learning, since our assumptions do admit polynomial sample complexity upper bounds if the data distribution is fixed.

- Our lower bounds are not pathological in nature and suggest that these concerns may arise in practice. In a precise sense, almost all feature extractors induce a hard MDP instance in our construction (see Section 4.4).

Instead, one interpretation is that the hardness is due to a distribution mismatch in the following sense: the agent does not know which distribution to use for minimizing a (supervised) learning error (see Kakade (2003) for discussion), and even a known transition model is not information-theoretically sufficient to reduce the sample complexity.

Furthermore, our work implies several interesting exponential separations on the sample complexity between: 1) value-based learning with perfect representation and value-based learning with a good-but-not-perfect representation, 2) value-based learning and policy-based learning, 3) policy-based learning and supervised learning and 4) reinforcement learning and imitation learning. We provide more details in Section 5.

## 2    RELATED WORK

A summary of previous upper bounds, together with lower bounds proved in this paper, is provided in Table 1. Some key assumptions are formally stated in Section 3 and Section 4. Our lower bounds highlight that classical complexity measures in supervised learning including small approximation error and margin, and standard assumptions in reinforcement learning including optimality gap and deterministic systems, are not enough for efficient RL with function approximation. We need additional assumptions, e.g., ones used in previous upper bounds, for efficient RL.

### 2.1    PREVIOUS LOWER BOUNDS

Existing exponential lower bounds, to our knowledge, construct *unstructured* MDPs with an exponentially large state space and reduce a bandit problem with exponentially many arms to an MDP (Krishnamurthy et al., 2016; Sun et al., 2017). However, these lower bounds cannot apply to MDPs whose transition models, value functions, or policies can be approximated with some natural function classes, e.g., linear functions, neural networks, etc. The current paper gives the first set of lower bounds for RL with linear function approximation (and thus also hold for super classes of linear functions such as neural networks).

### 2.2    PREVIOUS UPPER BOUNDS

We divide previous algorithms (with provable guarantees) into three classes: those that utilize uncertainty-based bonuses (e.g. UCB variants or Thompson sampling variants); approximate dynamic programming variants (which often make assumptions with respect to concentrability coefficients); and direct policy search-based methods (such as conserve policy iteration (CPI, see Kakade (2003)) or policy gradient methods, which make assumptions with respect to distribution mismatch coefficients).

| Query Oracle | RL | Generative Model | Known Transition |
|---|---|---|---|
| Previous Upper Bounds | | | |
| Exact linear $Q^*$ + DetMDP (Wen & Van Roy, 2013) | ✓ | ✓ | ✓ |
| Exact linear $Q^*$ + Bellman-Rank (Jiang et al., 2017) | ✓ | ✓ | ✓ |
| Exact Linear $Q^*$ + Low Var + Gap (Du et al., 2019a) | ✓ | ✓ | ✓ |
| Exact Linear $Q^*$ + Gap (Open Problem / Theorem C.1) | ? | ✓ | ✓ |
| Exact Linear $Q^\pi$ for all $\pi$ (Open Problem / Theorem D.1) | ? | ✓ | ✓ |
| Approx. Linear $Q^\pi$ for all $\pi$ + Concentratability (Munos, 2005; Antos et al., 2008) | ✓✗ | ✓ | ✓ |
| Approx. Linear $Q^\pi$ for all $\pi$ + Bounded Dist Mismatch Coeff (Kakade & Langford, 2002) | ✓✗ | ✓ | ✓ |
| Lower Bounds (this work) | | | |
| Approx Linear $Q^*$ (Theorem 4.1) | × | × | × |
| Approx Linear $Q^\pi$ for all $\pi$ (Theorem 4.1) | × | × | × |
| $\ell_\infty$ Approx Linear MDP (Theorem 4.2) | × | × | × |
| Exact Linear $\pi^*$ + Margin + Gap + DetMDP (Theorem 4.3) | × | × | × |
| Exact Linear $Q^*$ (Open Problem) | ? | ? | ? |

Table 1: Summary of theoretical results on reinforcement learning with linear function approximation. See Section 2 for discussion on this table. RL, Generative Model, Known Transition are defined in Section 3.3. Exact linear $Q^*$: Assumption 4.1 with $\delta = 0$. Approx linear $Q^*$: Assumption 4.1 with $\delta = \Omega\left(\sqrt{\frac{H}{d}}\right)$. Exact linear $\pi^*$: Assumption 4.4. Margin: Assumption 4.5. Exact Linear $Q^\pi$ for all $\pi$: Assumption 4.2 with $\delta = 0$. Approximate Linear $Q^\pi$ for all $\pi$: Assumption 4.2 with $\delta = \Omega\left(\sqrt{\frac{H}{d}}\right)$. DetMDP: deterministic system defined in Section 3.1. Bellman-rank: Definition 5 in Jiang et al. (2017). Low Var: Assumption 1 in Du et al. (2019b). Gap: Assumption 3.1. Bounded Distribution Mismatch Coefficient: Definition 3.3 in Agarwal et al. (2019). $\ell_\infty$ Approx Linear MDP: Assumption 4.3 with $\delta = \Omega\left(\sqrt{\frac{H}{d}}\right)$. ✓: there exists an algorithm with polynomial sample complexity to find a near-optimal policy. ✓✗: requires certain condition on the initial distribution. ×: exponential number of samples is required. ?: open problem.

The first class of methods include those based on witness rank, Belman rank, and the Eluder dimension, while the latter two classes of algorithms make assumptions either on *concentrability coefficients* or on *distribution mismatch coefficients* (see Agarwal et al. (2019); Scherrer (2014) for discussions).

**Uncertainty bonus-based algorithms.** Now we discuss existing theoretical results on value-based learning with function approximation. The most relevant work is Wen & Van Roy (2013) which showed in deterministic systems, if the optimal $Q$-function is within a pre-specified function class which has bounded Eluder dimension, for which the class of linear functions is a special case, then the agent can learn the optimal policy using polynomial number of samples. This result has recently been generalized by Du et al. (2019a) which can deal with stochastic reward and low variance transition but requires strictly positive optimality gap. As we listed in Table 1, it is an open problem whether the condition that the optimal $Q$-function is linear itself is sufficient for efficient RL.

Li et al. (2011) proposed a $Q$-learning algorithm which requires the Know-What-It-Knows oracle. However, it is in general unknown how to implement such oracle in practice. Jiang et al. (2017) proposed the concept of Bellman Rank to characterize the sample complexity of value-based learning methods and gave an algorithm that has polynomial sample complexity in terms of the Bellman Rank, though the proposed algorithm is not computationally efficient. Bellman rank is bounded for a wide range of problems, including MDP with small number of hidden states, linear MDP, LQR, etc. Later work gave computationally efficient algorithms for certain special cases (Dann et al., 2018; Du et al., 2019a; Yang & Wang, 2019b; Jin et al., 2019). Recently, Witness rank, a generalization of Bellman rank to model-based methods, is studied in Sun et al. (2019).

**Approximate dynamic programming-based algorithms.** We now discuss approximate dynamic programming-based results characterized in terms of the concentrability coefficient. While classical approximate dynamic programming results typically require $\ell_\infty$-bounded errors, the notion of *concentrability* (originally due to (Munos, 2005)) permits sharper bounds in terms of average-case function approximation error, provided that the concentrability coefficient is bounded (e.g. see Munos (2005); Szepesvári & Munos (2005); Antos et al. (2008); Geist et al. (2019)). Under the assumption that this problem-dependent parameter is bounded, Munos (2005); Szepesvári & Munos (2005) and Antos et al. (2008) proved sample complexity and error bounds for approximate dynamic programming methods when there is a data collection policy (under which value-function fitting occurs) that induces a finite concentrability coefficient. The assumption that the concentrability coefficient is finite is in fact quite limiting. See Chen & Jiang (2019) which provides a more detailed discussion on this quantity.

**Direct policy search-based algorithms.** Stronger guarantees over approximate dynamic programming-based algrithm can be obtained with direct policy search-based methods, where instead of having a bounded concentrability coefficient, one only needs to have a bounded *distribution mismatch coefficient*. The latter assumption requires the agent to have access to a "good" initial state distribution (e.g. a measure which has coverage over where an optimal policy tends to visit); note that this assumption does not make restrictions over the class of MDPs. There are two classes of algorithms that fall into this category. First, there is Conservative Policy Iteration (Kakade & Langford, 2002), along with Policy Search by Dynamic Programming (PSDP) (Bagnell et al., 2004), and other boosting-style of policy search-based methods Scherrer & Geist (2014); Scherrer (2014), which have guarantees in terms of bounded distribution mismatch ratio. Second, more recently, Agarwal et al. (2019) showed that policy gradient styles of algorithms also have comparable guarantees.

**Recent extensions.** Subsequent to this work, the work by Van Roy & Dong (2019) and Lattimore & Szepesvari (2019) made notable contributions to the misspecified linear bandit problem. In particular, both papers found that Theorem 4.1 in our paper can be extended to the misspecified linear bandit problem and gave upper bounds for this problem showing that our lower bound has tight dependency on $\delta$ and $d$. Lattimore & Szepesvari (2019) further gave an upper bound for the setting where the $Q$-functions of all policies can be approximated by linear functions with small approximation errors and the agent can interact with the environment using a generative model. This upper bound also demonstrates that our lower bound has tight dependency on $\delta$ and $d$.

## 3 Preliminaries

Throughout this paper, for a given integer $H$, we use $[H]$ to denote the set $\{0, 1, \ldots, H-1\}$.

## 3.1 EPISODIC REINFORCEMENT LEARNING

Let $\mathcal{M} = (\mathcal{S}, \mathcal{A}, H, P, R)$ be an Markov Decision Process (MDP) where $\mathcal{S}$ is the state space, $\mathcal{A}$ is the action space whose size is bounded by a constant, $H \in \mathbb{Z}_+$ is the planning horizon, $P : \mathcal{S} \times \mathcal{A} \rightarrow \triangle(\mathcal{S})$ is the transition function which takes a state-action pair and returns a distribution over states and $R : \mathcal{S} \times \mathcal{A} \rightarrow \triangle(\mathbb{R})$ is the reward distribution. Without loss of generality, we assume a fixed initial state $s_0$[4]. A policy $\pi : \mathcal{S} \rightarrow \triangle(\mathcal{A})$ prescribes a distribution over actions for each state. The policy $\pi$ induces a (random) trajectory $s_0, a_0, r_0, s_1, a_1, r_1, \ldots, s_{H-1}, a_{H-1}, r_{H-1}$ where $a_0 \sim \pi(s_0)$, $r_0 \sim R(s_0, a_0)$, $s_1 \sim P(s_0, a_0)$, $a_1 \sim \pi(s_1)$, etc. To streamline our analysis, for each $h \in [H]$, we use $\mathcal{S}_h \subseteq \mathcal{S}$ to denote the set of states at level $h$, and we assume $\mathcal{S}_h$ do not intersect with each other. We also assume $\sum_{h=0}^{H-1} r_h \in [0, 1]$ almost surely. Our goal is to find a policy $\pi$ that maximizes the expected total reward $\mathbb{E}\left[\sum_{h=0}^{H-1} r_h \mid \pi\right]$. We use $\pi^*$ to denote the optimal policy. We say a policy $\pi$ is $\varepsilon$-optimal if $\mathbb{E}\left[\sum_{h=0}^{H-1} r_h \mid \pi\right] \geq \mathbb{E}\left[\sum_{h=0}^{H-1} r_h \mid \pi^*\right] - \varepsilon$.

In this paper we prove lower bounds for deterministic systems, i.e., MDPs with deterministic transition $P$, deterministic reward $R$. In this setting, $P$ and $R$ can be regarded as functions instead of distributions. Since deterministic systems are special cases of general stochastic MDPs, lower bounds proved in this paper still hold for more general MDPs.

## 3.2 $Q$-FUNCTION AND OPTIMALITY GAP

An important concept in RL is the $Q$-function. Given a policy $\pi$, a level $h \in [H]$ and a state-action pair $(s, a) \in \mathcal{S}_h \times \mathcal{A}$, the $Q$-function is defined as $Q_h^\pi(s, a) = \mathbb{E}\left[\sum_{h'=h}^{H-1} r_{h'} \mid s_h = s, a_h = a, \pi\right]$. For simplicity, we denote $Q_h^*(s, a) = Q_h^{\pi^*}(s, a)$. In addition to these definitions, we list below an important assumption, the optimality gap assumption, which is widely used in reinforcement learning and bandit literature. To state the assumption, we first define the function $\mathrm{gap} : \mathcal{S} \times \mathcal{A} \rightarrow \mathbb{R}$ as $\mathrm{gap}(s, a) = \arg\max_{a' \in \mathcal{A}} Q^*(s, a') - Q^*(s, a)$. Now we formally state the assumption.

**Assumption 3.1** (Optimality Gap). *There exists $\rho > 0$ such that $\rho \leq \mathrm{gap}(s, a)$ for all $(s, a) \in \mathcal{S} \times \mathcal{A}$ with $\mathrm{gap}(s, a) > 0$.*

Here, $\rho$ is the smallest reward-to-go difference between the best set of actions and the rest. Recently, Du et al. (2019b) gave a provably efficient $Q$-learning algorithm based on this assumption and Simchowitz & Jamieson (2019) showed that with this condition, the agent only incurs logarithmic regret in the tabular setting.

## 3.3 QUERY MODELS

Here we discuss three possible query oracles interacting with the MDP.

- RL: The most basic and weakest query oracle for MDP is the standard reinforcement learning query oracle where the agent can only interact with the MDP by choosing actions and observe the next state and the reward.

- Generative Model: A stronger query model assumes the agent can transit to any state (Kearns & Singh, 2002; Kakade, 2003; Sidford et al., 2018). This query model is available in certain robotic applications where one can control the robot to reach the target state.

- Known Transition: The strongest query model considered is that the agent can not only transit to any state, but also knows the whole transition function. In this model, only the reward is unknown.

In this paper, we will prove lower bounds for the strongest Known Transition query oracle. Therefore, our lower bounds also apply to RL and Generative Model query oracles.

---

[4]Some papers assume the initial state is sampled from a distribution $P_1$. Note this is equivalent to assuming a fixed initial state $s_0$, by setting $P(s_0, a) = P_1$ for all $a \in \mathcal{A}$ and now our state $s_1$ is equivalent to the initial state in their assumption.

## 4 MAIN RESULTS

In this section we formally present our lower bounds. We also discuss proof ideas in Section 4.4.

### 4.1 LOWER BOUND FOR VALUE-BASED LEARNING

We first present our lower bound for value-based learning. A common assumption is that the $Q$-function can be predicted well by a linear function of the given features (representation) (Bertsekas & Tsitsiklis, 1996). Formally, the agent is given a feature extractor $\phi : \mathcal{S} \times \mathcal{A} \to \mathbb{R}^d$ which can be hand-crafted or a pre-trained neural network that transforms a state-action pair to a $d$-dimensional embedding. The following assumption states that the given feature extractor can be used to predict the $Q$-function with approximation error at most $\delta$ using a linear function.

**Assumption 4.1.** *There exists $\delta > 0$ and $\theta_0, \theta_1, \ldots, \theta_{H-1} \in \mathbb{R}^d$ such that for any $h \in [H]$ and any $(s, a) \in \mathcal{S}_h \times \mathcal{A}$, $|Q_h^*(s, a) - \langle \theta_h, \phi(s, a) \rangle| \leq \delta$.*

Here $\delta$ is the approximation error, which indicates the quality of the representation. If $\delta = 0$, then $Q$-function can be perfectly predicted by a linear function of $\phi(\cdot, \cdot)$. In general, $\delta$ becomes smaller as we increase the dimension of $\phi$, since larger dimension usually has more expressive power. When the feature extractor is strong enough, previous papers (Chen & Jiang, 2019; Farahmand, 2011) assume that linear functions of $\phi$ can approximate the $Q$-function of *any* policy.

**Assumption 4.2** (Policy Completeness). *There exists $\delta > 0$, such that for any $h \in [H]$ and any policy $\pi$, there exists $\theta_h^\pi \in \mathbb{R}^d$ such that for any $(s, a) \in \mathcal{S}_h \times \mathcal{A}$, $|Q_h^\pi(s, a) - \langle \theta_h, \phi(s, a) \rangle| \leq \delta$.*

In the theoretical reinforcement learning literature, Assumption 4.2 is often called the (approximate) policy completeness assumption. This assumption is crucial in proving polynomial sample complexity guarantee for value iteration type of algorithms (Chen & Jiang, 2019; Farahmand, 2011).

The following theorem shows when $\delta = \Omega\left(\sqrt{\frac{H}{d}}\right)$, the agent needs to sample exponential number of trajectories to find a near-optimal policy.

**Theorem 4.1** (Exponential Lower Bound for Value-based Learning). *There exists a family of MDPs with $|\mathcal{A}| = 2$ and a feature extractor $\phi$ that satisfy Assumption 4.2, such that any algorithm that returns a $1/2$-optimal policy with probability $0.9$ needs to sample $\Omega\left(\min\{|\mathcal{S}|, 2^H, \exp(d\delta^2/16)\}\right)$ trajectories.*

Note this lower bound also applies to MDPs that satisfy Assumption 4.1, since Assumption 4.2 is strictly stronger. We would like to emphasize that since linear functions is a subclass of more complicated function classes, e.g., neural networks, our lower bound also holds for these function classes. Moreover, in many scenarios, the feature extractor $\phi$ is the last layer of a neural network. Modern neural networks are often over-parameterized, which makes $d$ large. In this case, $d$ is much larger than $H$. Thus, our lower bound holds even if the representation has small approximation error. Furthermore, the assumption that $|\mathcal{A}| = 2$ is only for simplicity. Our lower bound can be easily generalized to the case that $|\mathcal{A}| > 2$, in which case the sample complexity lower bound is $\Omega\left(\min\{|\mathcal{S}|, |\mathcal{A}|^H, \exp(d\delta^2/16)\}\right)$.

### 4.2 LOWER BOUND FOR MODEL-BASED LEARNING

Here we present our lower bound for model-based learning. Recently, Yang & Wang (2019b) proposed the linear transition assumption which was later studied in Yang & Wang (2019a); Jin et al. (2019). Again, we assume the agent is given a feature extractor $\phi : \mathcal{S} \times \mathcal{A} \to \mathbb{R}^d$, and now we state the assumption formally as follow.

**Assumption 4.3** (Approximate Linear MDP). *There exists $\delta > 0$, $\beta_0, \beta_1, \ldots, \beta_{H-1} \in \mathbb{R}^d$ and $\psi : \mathcal{S} \to \mathbb{R}^d$ such that for any $h \in [H-1]$, $(s, a) \in \mathcal{S}_h \times \mathcal{A}$ and $s' \in \mathcal{S}_{h+1}$, $|P(s' \mid s, a) - \langle \psi(s'), \phi(s, a) \rangle| \leq \delta$ and $|\mathbb{E}[R(s, a)] - \langle \beta_h, \phi(s, a) \rangle| \leq \delta$.*

It has been shown in Yang & Wang (2019b;a); Jin et al. (2019) if $\|P(\cdot \mid s, a) - \langle \psi(\cdot), \phi(s, a) \rangle\|_1$ is bounded, then the problem admits an algorithm with polynomial sample complexity. Now we show that when $\delta = \Omega\left(\sqrt{\frac{H}{d}}\right)$ in Assumption 4.3, the agent needs exponential number of samples to find a near-optimal policy.

**Theorem 4.2** (Exponential Lower Bound for Linear Transition Model). *There exists a family of MDPs with $|\mathcal{A}| = 2$ and a feature extractor $\phi$ that satisfy Assumption 4.3, such that any algorithm that returns a $1/2$-optimal policy with probability $0.9$ needs to sample $\Omega\left(\min\{|\mathcal{S}|, 2^H, \exp(d\delta^2/16)\}\right)$ trajectories.*

Again, our lower bound can be easily generalized to the case that $|\mathcal{A}| > 2$.

We do note that an $\ell_\infty$ approximation for a transition matrix may be a weak condition. Under the stronger condition that the transition matrix can be approximated well under the total variational distance, there exists polynomial sample complexity upper bounds that can tolerate approximation errors (Yang & Wang, 2019b;a; Jin et al., 2019).

### 4.3 Lower Bound for Policy-based Learning

Next we present our lower bound for policy-based learning. This class of methods use function approximation on the policy and use optimization techniques, e.g., policy gradient, to find the optimal policy. In this paper, we focus on linear policies on top of a given representation. A linear policy $\pi$ is a policy of the form $\pi(s_h) = \arg\max_{a \in \mathcal{A}} \langle \theta_h, \phi(s_h, a) \rangle$ where $s_h \in \mathcal{S}_h$, $\phi(\cdot, \cdot)$ is a given feature extractor and $\theta_h \in \mathbb{R}^d$ is the linear coefficient. Note that applying policy gradient on softmax parameterization of the policy is indeed trying to find the optimal policy among linear policies.

Similar to value-based learning, a natural assumption for policy-based learning is that the optimal policy is realizable[5], i.e., the optimal policy is linear.

**Assumption 4.4.** *For any $h \in [H]$, there exists $\theta_h \in \mathbb{R}^d$ that satisfies for any $s \in \mathcal{S}_h$, we have $\pi^*(s) \in \arg\max_a \langle \theta_h, \phi(s, a) \rangle$.*

Here we discuss another assumption. For learning a linear classifier in the supervised learning setting, one can reduce the sample complexity significantly if the optimal linear classifier has a margin.

**Assumption 4.5.** *We assume $\phi(s, a) \in \mathbb{R}^d$ satisfies $\|\phi(s, a)\|_2 = 1$ for any $(s, a) \in \mathcal{S} \times \mathcal{A}$. For any $h \in [H]$, there exists $\theta_h \in \mathbb{R}^d$ with $\|\theta_h\|_2 = 1$ and $\triangle > 0$ such that for any $s \in \mathcal{S}_h$, there is a unique optimal action $\pi^*(s)$, and for any $a \neq \pi^*(s)$, $\langle \theta_h, \phi(s, \pi^*(s)) \rangle - \langle \theta_h, \phi(s, a) \rangle \geq \triangle$.*

Here we restrict the linear coefficients and features to have unit norm for normalization. Note that Assumption 4.5 is strictly stronger than Assumption 4.4. Now we present our result for linear policy.

**Theorem 4.3** (Exponential Lower Bound for Policy-based Learning). *There exists an absolute constant $\triangle_0$, such that for any $\triangle \leq \triangle_0$, there exists a family of MDPs with $|\mathcal{A}| = 2$ and a feature extractor $\phi$ that satisfy Assumption 3.1 with $\rho = \frac{1}{2\min\{H,d\}}$ and Assumption 4.5, such that any algorithm that returns a $1/4$-optimal policy with probability at least $0.9$ needs to sample $\Omega\left(\min\{2^H, 2^d\}\right)$ trajectories.*

Again, our lower bound can be easily generalized to the case that $|\mathcal{A}| > 2$.

Compared with Theorem 4.1, Theorem 4.3 is even more pessimistic, in the sense that even with perfect representation with benign properties (gap and margin), the agent still needs to sample exponential number of samples. It also suggests that policy-based learning could be very different from supervised learning.

### 4.4 Proof Ideas

**The binary tree hard instance.** All our lower bound are proved based on reductions from the following hard instance. In this instance, both the transition $P$ and the reward $R$ are deterministic. There are $H$ levels of states, which form a full binary tree of depth $H$. There are $2^h$ states in level $h$, and thus $2^H - 1$ states in total. Among all the $2^{H-1}$ states in level $H - 1$, there is only one state with reward $R = 1$, and for all other states in the MDP, the corresponding reward value $R = 0$. Intuitively, to find a $1/2$-optimal policy for such MDPs, the agent must enumerate all possible states in level $H - 1$ to find the state with reward $R = 1$. Doing so intrinsically induces a sample complexity of $\Omega(2^H)$. This intuition is formalized in Theorem A.1 using Yao's minimax principle (Yao, 1977).

---

[5]Unlike value-based learning, it is hard to define completeness on the policy-based learning with function approximation, since not all policy has the $\arg\max$ form.

**Lower bound for value-based and model-based learning** We now show how to construct a set of features so that Assumption 4.1-4.3 hold. Our main idea is to the utilize the following fact regarding the identity matrix: $\varepsilon\text{-rank}(I_{2^H}) \leq O(H/\varepsilon^2)$. Here for a matrix $A \in \mathbb{R}^{n \times n}$, its $\varepsilon$-rank (a.k.a *approximate rank*) is defined to be $\min\{\text{rank}(B) : B \in \mathbb{R}^{n \times n}, \|A - B\|_\infty \leq \varepsilon\}$, where we use $\|\cdot\|_\infty$ to denote the entry-wise $\ell_\infty$ norm of a matrix. The upper bound $\varepsilon\text{-rank}(I_n) \leq O(\log n/\varepsilon^2)$ was first proved in Alon (2009) using the Johnson-Lindenstrauss Lemma (Johnson & Lindenstrauss, 1984), and we also provide a proof in Lemma A.1. The concept of $\varepsilon$-rank has wide applications in theoretical computer science (Alon, 2009; Barak et al., 2011; Alon et al., 2013; 2014; Chen & Wang, 2019), but to our knowledge, this is the first time that it appears in reinforcement learning.

This fact can be alternatively stated as follow: there exists $\Phi \in \mathbb{R}^{2^H \times O(H/\varepsilon^2)}$ such that $\|I_{2^H} - \Phi\Phi^\top\|_\infty \leq \varepsilon$. We interpret each row of $\Phi$ as the feature of a state in the binary tree. By construction of $\Phi$, now features of states in the binary tree have a nice property that (i) each feature vector has approximately unit norm and (ii) different feature vector are nearly orthogonal. Using this set of features, we can now show that Assumption 4.1-4.3 hold. Here we prove Assumption 4.1 holds as an example and prove other assumptions also hold in the appendix. To prove Assumption 4.1, we note that in the binary tree hard instance, for each level $h$, only a single state satisfies $Q^* = 1$, and all other states satisfy $Q^* = 0$. We simply take $\theta_h$ to be the feature of the state with $Q^* = 1$. Since all feature vectors are nearly orthogonal, Assumption 4.1 holds.

Since the above fact regarding the $\varepsilon$-rank of the identity matrix can be proved by simply taking each row of $\Phi$ to be a random unit vector, our lower bound reveals another intriguing (yet pessimistic) aspect of Assumption 4.1-4.3: for the binary tree instance, almost all feature extractors induce a hard MDP instance. This again suggests that a good representation itself may not necessarily lead to efficient RL and additional assumptions (e.g. on the reward distribution) could be crucial.

**Lower bound for policy-based learning.** It is straightfoward to construct a set of feature vectors for the binary tree instance so that Assumption 4.4 holds, even if $d = 1$. We set $\phi(s, a)$ to be $+1$ if $a = a_1$ and $-1$ if $a = a_2$. For each level $h$, for the unique state $s$ in level $h$ with $Q^* = 1$, we set $\theta_h$ to be 1 if $\pi^*(s) = a_1$ and $-1$ if $\pi^*(s) = a_2$. With this construction, Assumption 4.4 holds.

To prove that the lower bound under Assumption 4.5, we use a new reward function for states in level $H - 1$ in the binary tree instance above so that there exists a unique optimal action for each state in the MDP. See Figure 2 for an example with $H = 3$ levels of states. Another nice property of the new reward function is that for all states $s$ we always have $\pi^*(s) = a_1$. Now, we define $2^{H-1}$ different new MDPs as follow: for each state in level $H - 1$, we change its original reward (defined in Figure 2) to 1. An exponential sample complexity lower bound for these MDPs can be proved using the same argument as the original binary tree hard instance, and now we show this set of MDPs satisfy Assumption 4.5. We first show in Lemma A.2 that there exists a set $\mathcal{N} \subseteq \mathbb{S}^{d-1}$ with $|\mathcal{N}| = (1/\triangle)^{\Omega(d)}$, so that for each $p \in \mathcal{N}$, there exists a hyperplane $L$ that separates $p$ and $\mathcal{N} \setminus \{p\}$, and all vectors in $\mathcal{N}$ have distance at least $\triangle$ to $L$. Equivalently, for each $p \in \mathcal{N}$, we can always define a linear function $f_p$ so that $f_p(p) \geq \triangle$ and $f_p(q) \leq -\triangle$ for all $q \in \mathcal{N} \setminus \{p\}$. This can be proved using standard lower bounds on the size of $\varepsilon$-nets. Now we simply use vectors in $\mathcal{N}$ as features of states. By construction of the reward function, for each level $h$, there could only be two possible cases for the optimal policy $\pi^*$. I.e., either $\pi^*(s) = a_1$ for all states in level $h$, or $\pi^*(s) = a_2$ for a unique state $s$ and $\pi^*(s') = a_1$ for all $s \neq s'$. In both cases, we can easily define a linear function with margin $\triangle$ to implement the optimal policy $\pi^*$, and thus Assumption 4.5 holds. Notice that in this proof, we critically relies on $d = \Theta(H)$, so that we can utilize the curse of dimensionality to construct a large set of vectors as features.

## 5 SEPARATIONS

**Perfect representation vs. good-but-not-perfect representation.** For value-based learning in deterministic systems, Wen & Van Roy (2013) showed polynomial sample complexity upper bound when the representation can perfectly predict the $Q$-function. In contrast, if the representation is only able to *approximate* the $Q$-function, then the agent requires exponential number of trajectories. This exponential separation demonstrates a *provable exponential benefit of better representation.*

**Value-based learning vs. policy-based learning.** Note that if the optimal $Q$-function can be perfectly predicted by the provided representation, then the optimal policy can also be perfectly

predicted using the same representation. Since Wen & Van Roy (2013) showed polynomial sample complexity upper bound when the representation can perfectly predict the $Q$-function, our lower bound on policy-based learning, which applies to perfect representations, thus demonstrates that *the ability of predicting the Q-function is much stronger than that of predicting the optimal policy.*

**Supervised learning vs. reinforcement learning.** For policy-based learning, if the planning horizon $H = 1$, the problem becomes learning a linear classifier, for which there are polynomial sample complexity upper bounds. For policy-based learning, the agent needs to learn $H$ linear classifiers sequentially. Our lower bound on policy-based learning shows the sample complexity dependency on $H$ is exponential.

**Imitation learning vs. reinforcement learning.** In imitation learning (IL), the agent can observe trajectories induced by the optimal policy (expert). If the optimal policy is linear in the given representation, it can be shown that the simple behavior cloning algorithm only requires polynomial number of samples to find a near-optimal policy (Ross et al., 2011). Our Theorem 4.3 shows if the agent cannot observe expert's behavior, then it requires exponential number of samples. Therefore, our lower bound shows there is an *exponential separation between policy-based RL and IL* when function approximation is used.

## 6   ACKNOWLEDGMENTS

The authors would like to thank Yuping Luo, Wenlong Mou, Martin Wainwright, Mengdi Wang and Yifan Wu for insightful discussions. Also, the authors would also like to gratefully acknowledge Benjamin Van Roy, Shi Dong, Tor Lattimore and Csaba Szepesvári for sharing a draft of their work and their comments. Simon S. Du is supported by NSF grant DMS-1638352 and the Infosys Membership. Sham M. Kakade acknowledges funding from the Washington Research Foundation Fund for Innovation in Data-Intensive Discovery; the NSF award CCF 1740551; and the ONR award N00014-18-1-2247. Ruosong Wang is supported in part by NSF IIS1763562, AFRL CogDeCON FA875018C0014, and DARPA SAGAMORE HR00111990016. Part of this work was done while Simon S. Du was visiting Google Brain Princeton and Ruosong Wang was visiting Princeton University.

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

# A    Proofs of Lower Bounds

In this section we present our lower bounds. It will also be useful to define the value function of a given state $s \in \mathcal{S}_h$ as $V_h^\pi(s) = \mathbb{E}\left[\sum_{h'=h}^{H-1} r_{h'} \mid s_h = s, \pi\right]$. For simplicity, we denote $V_h^* = V_h^{\pi^*}(s)$. Throughout the appendix, for the $Q$-function $Q_h^\pi$ and $Q_h^*$ and the value function $V_h^\pi$ and $V_h^*$, we may omit $h$ from the subscript when it is clear from the context.

We first introduce the INDEX-QUERY problem, which will be useful in our lower bound arguments.

**Definition A.1** (INDEX-QUERY). *In the $\mathsf{INDQ}_n$ problem, there is an underlying integer $i^* \in [n]$. The algorithm sequentially (and adaptively) outputs guesses $i \in [n]$ and queries whether $i = i^*$. The goal is to output $i^*$, using as few queries as possible.*

**Definition A.2** ($\delta$-correct algorithms). *For a real number $\delta \in (0, 1)$, we say a randomized algorithm $\mathcal{A}$ is $\delta$-correct for $\mathsf{INDQ}_n$, if for any underlying integer $i^* \in [n]$, with probability at least $1 - \delta$, $\mathcal{A}$ outputs $i^*$.*

The following theorem states the query complexity of $\mathsf{INDQ}_n$ for $0.1$-correct algorithms, whose proof is provided in Section B.1.

**Theorem A.1.** *Any $0.1$-correct algorithm $\mathcal{A}$ for $\mathsf{INDQ}_n$ requires at least $0.9n$ queries in the worst case.*

## A.1    Proof of Lower Bound for Value-based Learning

In this section we prove Theorem 4.1. We need the following existential result, whose proof is provided in Section B.2.

**Lemma A.1.** *For any $n > 2$, there exists a set of vectors $\mathcal{P} = \{p_0, p_1, \dots, p_{n-1}\} \subset \mathbb{R}^d$ with $d = \lceil 8 \ln n / \varepsilon^2 \rceil$ such that*

1. *$\|p_i\|_2 = 1$ for all $0 \le i \le n - 1$;*

2. *$|\langle p_i, p_j \rangle| \le \varepsilon$ for any $0 \le i, j \le n - 1$ with $i \ne j$.*

Now we give the construction of the hard MDP instances. We first define the transitions and the reward functions. In the hard instances, both the rewards and the transitions are deterministic. There are $H$ levels of states, and level $h \in [H]$ contains $2^h$ distinct states. Thus we have $|\mathcal{S}| = 2^H - 1$. If $|\mathcal{S}| > 2^H - 1$ we simply add dummy states to the state space $\mathcal{S}$. We use $s_0, s_1, \dots, s_{2^H-2}$ to name these states. Here, $s_0$ is the unique state in level $h = 0$, $s_1$ and $s_2$ are the two states in level $h = 1$, $s_3, s_4, s_5$ and $s_6$ are the four states in level $h = 2$, etc. There are two different actions, $a_1$ and $a_2$, in the MDPs. For a state $s_i$ in level $h$ with $h < H - 1$, playing action $a_1$ transits state $s_i$ to state $s_{2i+1}$ and playing action $a_2$ transits state $s_i$ to state $s_{2i+2}$, where $s_{2i+1}$ and $s_{2i+2}$ are both states in level $h + 1$. See Figure 1 for an example with $H = 3$.

In our hard instances, $r(s, a) = 0$ for all $(s, a)$ pairs except for a unique state $s$ in level $H - 2$ and a unique action $a \in \{a_1, a_2\}$. It is convenient to define $\overline{r}(s') = r(s, a)$, if playing action $a$ transits $s$ to $s'$. For our hard instances, we have $\overline{r}(s) = 1$ for a unique node $s$ in level $H - 1$ and $\overline{r}(s) = 0$ for all other nodes.

Now we define the features map $\phi(\cdot, \cdot)$. Here we assume $d \ge 2 \cdot \lceil 8 \ln 2 \cdot H / \delta^2 \rceil$, and otherwise we can simply decrease the planning horizon so that $d \ge 2 \cdot \lceil 8 \ln 2 \cdot H / \delta^2 \rceil$. We invoke Lemma A.1 to get a set $\mathcal{P} = \{p_0, p_1, \dots, p_{2^H-1}\} \subset \mathbb{R}^{d/2}$. For each state $s_i$, $\phi(s_i, a_1) \in \mathbb{R}^d$ is defined to be $[p_i; 0]$, and $\phi(s_i, a_2) \in \mathbb{R}^d$ is defined to be $[0; p_i]$. This finishes the definition of the MDPs. We now show that no matter which state $s$ in level $H - 1$ satisfies $\overline{r}(s) = 1$, the resulting MDP always satisfies Assumption 4.2.

**Verifying Assumption 4.2.**    By construction, for each level $h \in [H]$, there is a unique state $s_h$ in level $h$ and action $a_h \in \{a_1, a_2\}$, such that $Q^*(s_h, a_h) = 1$. For all other $(s, a)$ pairs such that $s \ne s_h$ or $a \ne a_h$, it is satisfied that $Q^*(s, a) = 0$. For a given level $h$ and policy $\pi$, we take $\theta_h^\pi$ to be $Q^\pi(s_h, a_h) \cdot \phi(s_h, a_h)$. Now we show that $|Q^\pi(s, a) - \langle \theta_h^\pi, \phi(s, a) \rangle| \le \delta$ for all states $s$ in level $h$ and $a \in \{a_1, a_2\}$.

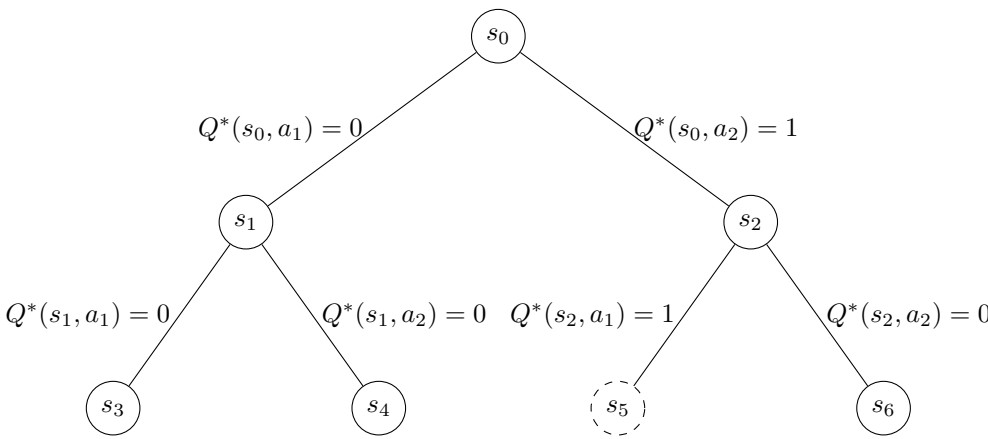

Figure 1: An example with $H = 3$. For this example, we have $\overline{r}(s_5) = 1$ and $\overline{r}(s) = 0$ for all other states $s$. The unique state $s_5$ which satisfies $\overline{r}(s) = 1$ is marked as dash in the figure. The induced $Q^*$ function is marked on the edges.

**Case I:** $a \neq a_h$**.** In this case, we have $Q^\pi(s, a) = 0$ and $\langle \theta_h^\pi, \phi(s, a) \rangle = 0$, since $\theta_h^\pi$ and $\phi(s, a)$ do not have a common non-zero coordinate.

**Case II:** $a = a_h$ **and** $s \neq s_h$**.** In this case, by the second property of $\mathcal{P}$ in Lemma A.1 and the fact that $Q^\pi(s_h, a_h) \leq 1$, we have $|\langle \theta_h^\pi, \phi(s, a) \rangle| \leq \delta$. Meanwhile, we have $Q^\pi(s, a) = 0$.

**Case III:** $a = a_h$ **and** $s = s_h$**.** In this case, we have $\langle \theta_h^\pi, \phi(s, a) \rangle = Q^\pi(s_h, a_h)$.

Finally, we prove any algorithm that solves these MDP instances and succeeds with probability at least 0.9 needs to sample at least $\frac{9}{20} \cdot 2^H$ trajectories. We do so by providing a reduction from $\mathsf{INDQ}_{2^{H-1}}$ to solving MDPs. Suppose we have an algorithm for solving these MDPs, we show that such an algorithm can be transformed to solve $\mathsf{INDQ}_{2^{H-1}}$. For a specific choice of $i^*$ in $\mathsf{INDQ}_{2^{H-1}}$, there is a corresponding MDP instance with

$$\overline{r}(s) = \begin{cases} 1 & \text{if } s = s_{i^* + 2^{H-1} - 1} \\ 0 & \text{otherwise} \end{cases}.$$

Notice that for all MDPs that we are considering, the transition and features are always the same. Thus, the only thing that the learner needs to learn by interacting with the environment is the reward value. Since the reward value is non-zero only for states in level $H - 1$, each time the algorithm for solving MDP samples a trajectory that ends at state $s_i$ where $s_i$ is a state in level $H - 1$, we query whether $i^* = i - 2^{H-1} + 1$ or not in $\mathsf{INDQ}_{2^{H-1}}$, and return reward value 1 if $i^* = i - 2^{H-1} + 1$ and 0 otherwise. If the algorithm is guaranteed to return a $1/2$-optimal policy, then it must be able to find $i^*$.

## A.2 PROOF OF LOWER BOUND FOR MODEL-BASED LEARNING

*Proof of Theorem 4.2.* We use the same construction as in the proof of Theorem 4.1. Note we just need to verify that the construction satisfies Assumption 4.3. By construction, for all $h \in \{1, 2, \ldots, H - 1\}$, for each state $s'$ in level $h$, there exists a unique $(s, a)$ pair such that playing action $a$ transits $s$ to $s'$, and we take $\psi(s') = \phi(s, a)$. We also take $\beta_h = 0$ for $h \in \{0, 1, \ldots, H - 4, H - 3\}$ and $\beta_{H-2} = \phi(s, a)$ where $(s, a)$ is the unique pair with $R(s, a) = 1$. Now, according to the design of $\phi(\cdot, \cdot)$ and Lemma A.1, Assumption 4.3 is satisfied. $\qquad\square$

## A.3 PROOF OF LOWER BOUND FOR POLICY-BASED LEARNING

In this section, we present our hardness results for linear policy learning. We first prove a weaker lower bound which only satisfies Assumption 4.4, and then prove Theoerem 4.3.

**Warmup: Lower Bound for Linear Policy Without Margin.** To present the hardness results, we first give the construction of the hard instances. The transitions and rewards functions of these MDP instances are exactly the same as those in Section A.1. The main difference is in the definition of the feature map $\phi(\cdot, \cdot)$. For this lower bound, we define $\phi(s, a) = 1 \in \mathbb{R}$ if $a = a_1$ and $\phi(s, a) = -1$ if $a = a_2$. By construction, these MDPs satisfy Assumption 3.1 with $\rho = 1$. We now show that no matter which state $s$ in level $H - 1$ satisfies $\bar{r}(s) = 1^6$, the resulting MDP always satisfies Assumption 4.4.

**Verifying Assumption 4.4.** Recall that for each level $h \in [H]$, there is a unique state $s_h$ in level $h$ and action $a_h \in \{a_1, a_2\}$, such that $Q^*(s_h, a_h) = 1$. For all other $(s, a)$ pairs such that $s \neq s_h$ or $a \neq a_h$, it is satisfied that $Q^*(s, a) = 0$. We simply take $\theta_h$ to be $1$ if $a_h = a_1$, and take $\theta_h$ to be $-1$ if $a_h = a_2$.

Using the same lower bound argument (by reducing INDEX-QUERY to MDPs), we have the following theorem.

**Theorem A.2.** *There exists a family of MDPs and a feature map $\phi(\cdot, \cdot)$ that satisfy Assumption 4.4 with $d = 1$ and Assumption 3.1 with $\rho = 1$, such that any algorithm that returns a $1/2$-optimal policy with probability at least $0.9$ needs to sample $\Omega\left(2^H\right)$ trajectories.*

**Proof of Theoerem 4.3** Now we prove Theoerem 4.3. In order to prove Theoerem 4.3, we need the following geometric lemma whose proof is provided in Section B.3.

**Lemma A.2.** *Let $d \in \mathbb{N}_+$ be a positive integer and $\epsilon \in (0, 1)$ be a real number. Then there exists a set of points $\mathcal{N} \subset \mathbb{S}^{d-1}$ with size $|\mathcal{N}| = \Omega(1/\epsilon^{d/2})$ such that for every point $x \in \mathcal{N}$,*

$$\inf_{y \in \text{conv}(\mathcal{N} \setminus \{x\})} \|x - y\|_2 \geq \epsilon/2. \tag{1}$$

Now we are ready to prove Theorem 4.3. In the proof we assume $H = d$, since otherwise we can take $H$ and $d$ to be $\min\{H, d\}$ by decreasing the planning horizon $H$ or adding dummy dimensions to the feature extractor $\phi$.

*Proof of Theorem 4.3.* We define a set of $2^{H-1}$ deterministic MDPs. The transitions of these hard instances are exactly the same as those in Section A.1. The main difference is in the definition of the feature map $\phi(\cdot, \cdot)$ and the reward function. Again in the hard instances, $r(s, a) = 0$ for all $s$ in the first $H - 2$ levels. Using the terminology in Section A.1, we have $\bar{r}(s) = 0$ for all states in the first $H - 1$ levels. Now we define $\bar{r}(s)$ for states $s$ in level $H - 1$. We do so by recursively defining the optimal value function $V^*(\cdot)$. The initial state $s_0$ in level $0$ satisfies $V^*(s_0) = 1/2$. For each state $s_i$ in the first $H - 2$ levels, we have $V^*(s_{2i+1}) = V^*(s_i)$ and $V^*(s_{2i+2}) = V^*(s_i) - 1/2H$. For each state $s_i$ in the level $h = H - 2$, we have $\bar{r}(s_{2i+1}) = V^*(s_i)$ and $\bar{r}(s_{2i+2}) = V^*(s_i) - 1/2H$. This implies that $\rho = 1/2H$. In fact, this implies a stronger property that each state has a unique optimal action. See Figure 2 for an example with $H = 3$.

To define $2^{H-1}$ different MDPs, for each state $s$ in level $H - 1$ of the MDP defined above, we define a new MDP by changing $\bar{r}(s)$ from its original value to $1$. This also affects the definition of the optimal $V$ function for states in the first $H - 1$ levels. In particular, for each level $i \in \{0, 1, 2, \ldots, H - 2\}$, we have changed the $V$ value of a unique state in level $i$ from its original value (at most $1/2$) to $1$. By doing so we have defined $2^{H-1}$ different MDPs. See Figure 3 for an example with $H = 3$.

Now we define the feature function $\phi(\cdot, \cdot)$. We invoke Lemma A.2 with $\epsilon = 8\triangle$ and $d = H/2 - 1$. Since $\triangle$ is sufficiently small, we have $|\mathcal{N}| \geq 2^H$. We use $\mathcal{P} = \{p_0, p_2, \ldots, p_{2^H-1}\} \subset \mathbb{R}^{H/2-1}$ to denote an arbitrary subset of $\mathcal{N}$ with cardinality $2^H$. By Lemma A.2, for any $p \in \mathcal{P}$, the distance between $p$ and the convex hull of $\mathcal{P} \setminus \{p\}$ is at least $4\triangle$. Thus, there exists a hyperplane $L$ which separates $p$ and $\mathcal{P} \setminus \{p\}$, and for all points $q \in \mathcal{P}$, the distance between $q$ and $L$ is at least $2\triangle$. Equivalently, for each point $p \in \mathcal{P}$, there exists $n_p \in \mathbb{R}^{H/2-1}$ and $o_p \in \mathbb{R}$ such that $\|n_p\|_2 = 1$, $|o_p| \leq 1$ and the linear function $f_p(q) = \langle q, n_p \rangle + o_p$ satisfies $f_p(p) \geq 2\triangle$ and $f_p(q) \leq -2\triangle$ for all $q \in \mathcal{P} \setminus \{p\}$. Given the set $\mathcal{P} = \{p_0, p_2, \ldots, p_{2^H-1}\} \subset \mathbb{R}^{H/2-1}$, we construct a new set

---

[6]Recall that $\bar{r}(s') = r(s, a)$, if playing action $a$ transits $s$ to $s'$. Moreover, for the instances in Section A.1, we have $\bar{r}(s) = 1$ for a unique node $s$ in level $H - 1$ and $\bar{r}(s) = 0$ for all other nodes.

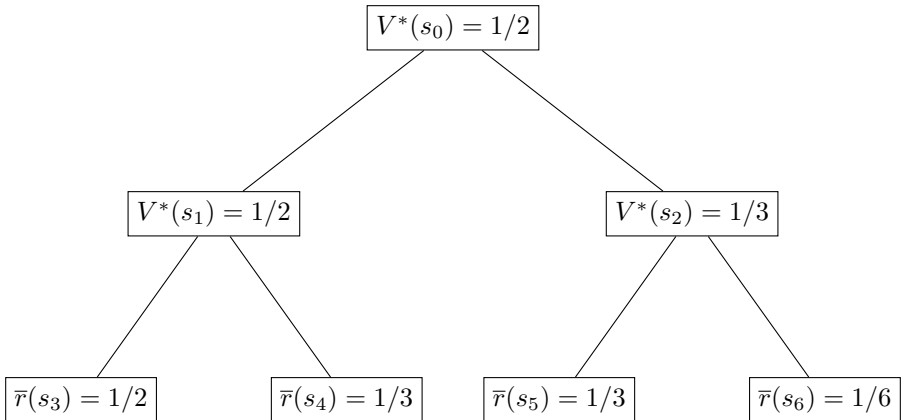

Figure 2: An example with $H = 3$.

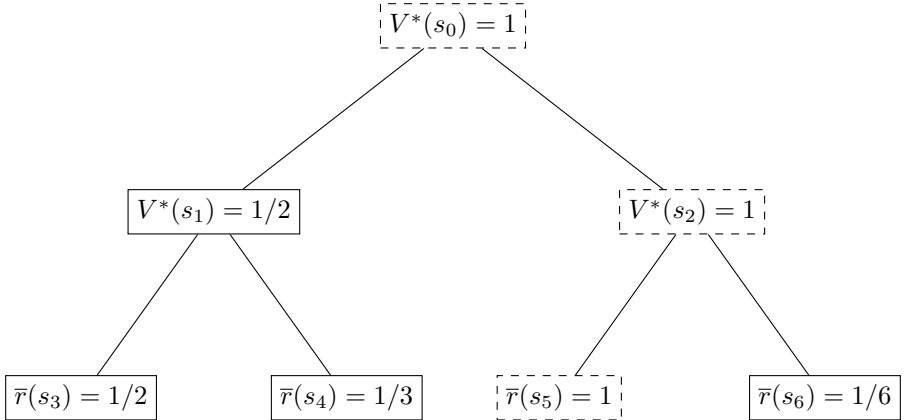

Figure 3: An example with $H = 3$. Here we define a new MDP by changing $\overline{r}(s_5)$ from its original value $1/3$ to 1. This also affects the value of $V(s_2)$ and $V(s_0)$.

$\overline{\mathcal{P}} = \{\overline{p}_0, \overline{p}_2, \ldots, \overline{p}_{2^H-1}\} \subset \mathbb{R}^{H/2}$, where $\overline{p}_i = [p_i; 1] \in \mathbb{R}^{H/2}$. Thus $\|\overline{p}_i\|_2 = \sqrt{2}$ for all $\overline{p}_i \in \overline{\mathcal{P}}$. Clearly, for each $\overline{p} \in \overline{\mathcal{P}}$, there exists a vector $\omega_{\overline{p}} \in \mathbb{R}^{H/2}$ such that $\langle \omega_{\overline{p}}, \overline{p} \rangle \geq 2\triangle$ and $\langle \omega_{\overline{p}}, \overline{q} \rangle \leq -2\triangle$ for all $\overline{q} \in \overline{\mathcal{P}} \setminus \{\overline{p}\}$. It is also clear that $\|\omega_{\overline{p}}\|_2 \leq \sqrt{2}$. We take $\phi(s_i, a_1) = [0; \overline{p}_i] \in \mathbb{R}^H$ and $\phi(s_i, a_2) = [\overline{p}_i; 0] \in \mathbb{R}^H$.

We now show that all the $2^{H-1}$ MDPs constructed above satisfy the linear policy assumption. Namely, we show that for any state $s$ in level $H - 1$, after changing $\overline{r}(s)$ to be 1, the resulting MDP satisfies the linear policy assumption. As in Section A.1, for each level $h \in [H]$, there is a unique state $s_h$ in level $h$ and action $a_h \in \{a_1, a_2\}$, such that $Q^*(s_h, a_h) = 1$. For all other $(s, a)$ pairs such that $s \neq s_h$ or $a \neq a_h$, it is satisfied that $Q^*(s, a) = 0$. For each level $h$, if $a_h = a_1$, then we take $(\theta_h)_{H/2} = 1$ and $(\theta_h)_H = -1$, and all other entries in $\theta_h$ are zeros. If $a_h = a_2$, we use $\overline{p}$ to denote the vector formed by the first $H/2$ coordinates of $\phi(s_h, a_2)$. By construction, we have $\overline{p} \in \overline{\mathcal{P}}$. We take $\theta_h = [\omega_{\overline{p}}; 0]$ in this case. In any case, we have $\|\theta_h\|_2 \leq \sqrt{2}$. Now for each level $h$, if $a_h = a_1$, then for all states $s$ in level $h$, we have $\pi^*(s) = a_1$. In this case, $\langle \phi(s, a_1), \theta_h \rangle = 1$ and $\langle \phi(s, a_2), \theta_h \rangle = -1$ for all states in level $h$, and thus Assumption 4.5 is satisfied. If $a_h = a_2$, then $\pi^*(s_h) = a_2$ and $\pi^*(s) = a_1$ for all states $s \neq s_h$ in level $h$. By construction, we have $\langle \theta_h, \phi(s, a_1) \rangle = 0$ for all states $s$ in level $h$, since $\theta_h$ and $\phi(s, a_1)$ do not have a common non-zero entry. We also have $\langle \theta_h, \phi(s_h, a_2) \rangle \geq 2\triangle$ and $\langle \theta_h, \phi(s, a_2) \rangle \leq -2\triangle$ for all states $s \neq s_h$ in level $h$. Finally, we normalize all $\theta_h$ and $\phi(s, a)$ so that they all have unit norm. Since $\|\phi(s, a)\|_2 = \sqrt{2}$ for all $(s, a)$ pairs before normalization, Assumption 4.5 is still satisfied after normalization.

Finally, we prove any algorithm that solves these MDP instances and succeeds with probability at least 0.9 needs to sample at least $\Omega(2^H)$ trajectories. We do so by providing a reduction from $\mathsf{INDQ}_{2^{H-1}}$ to solving MDPs. Suppose we have an algorithm for solving these MDPs, we show that such an algorithm can be transformed to solve $\mathsf{INDQ}_{2^{H-1}}$. For a specific choice of $i^*$ in $\mathsf{INDQ}_{2^{H-1}}$, there is a corresponding MDP instance with

$$\overline{r}(s) = \begin{cases} 1 & \text{if } s = s_{i^*+2^{H-1}-1} \\ \text{the original (recursively defined) value} & \text{otherwise} \end{cases}.$$

Notice that for all MDPs that we are considering, the transition and features are always the same. Thus, the only thing that the learner needs to learn by interacting with the environment is the reward value. Since the reward value is non-zero only for states in level $H - 1$, each time the algorithm for solving MDP samples a trajectory that ends at state $s_i$ where $s_i$ is a state in level $H - 1$, we query whether $i^* = i - 2^{H-1} + 1$ or not in $\mathsf{INDQ}_{2^{H-1}}$, and return reward value 1 if $i^* = i - 2^{H-1} + 1$ and it original reward value otherwise. If the algorithm is guaranteed to return a $1/4$-optimal policy, then it must be able to find $i^*$.

$\square$

# B  TECHNICAL PROOFS

## B.1  PROOF OF THEOREM A.1

*Proof.* The proof is a straightforward application of Yao's minimax principle Yao (1977). We provide the full proof for completeness.

Consider an input distribution where $i^*$ is drawn uniformly at random from $[n]$. Suppose there is a 0.1-correct algorithm for $\mathsf{INDQ}_n$ with worst-case query complexity $T$ such that $T < 0.9n$. By averaging, there is a deterministic algorithm $\mathcal{A}'$ with worst-case query complexity $T$, such that

$$\Pr_{i \sim [n]} [\mathcal{A}' \text{ correctly outputs } i \text{ when } i^* = i] \geq 0.9.$$

We may assume that the sequence of queries made by $\mathcal{A}'$ is fixed. This is because (i) $\mathcal{A}'$ is deterministic and (ii) before $\mathcal{A}'$ correctly guesses $i^*$, all responses that $\mathcal{A}'$ receives are the same (i.e., all guesses are incorrect). We use $S = \{s_1, s_2, \ldots, s_m\}$ to denote the sequence of queries made by $\mathcal{A}'$. Notice that $m$ is the worst-case query complexity of $\mathcal{A}'$. Suppose $m < 0.9n$, there exist $0.1n$ distinct $i \in [n]$ such that $\mathcal{A}'$ will never guess $i$, and will be incorrect if $i^*$ equals $i$, which implies

$$\Pr_{i \sim [n]} [\mathcal{A}' \text{ correctly outputs } i \text{ when } i^* = i] < 0.9.$$

$\square$

## B.2 Proof of Lemma A.1

We need the following tail inequality for random unit vectors, which will be useful for the proof of Lemma A.1.

**Lemma B.1** (Lemma 2.2 in Dasgupta & Gupta (2003)). *For a random unit vector $u$ in $\mathbb{R}^d$ and $\beta > 1$, we have*

$$\Pr\left[u_1^2 \geq \beta/d\right] \leq \exp((1 + \ln \beta - \beta)/2).$$

*In particular, when $\beta \geq 6$, we have*

$$\Pr\left[u_1^2 > \beta/d\right] \leq \exp(-\beta/4).$$

*Proof of Lemma A.1.* Let $\mathcal{Q} = \{q_1, q_2, \ldots, q_n\}$ be a set of $n$ independent random unit vectors in $\mathbb{R}^d$ with $d = \lceil 8 \ln n/\varepsilon^2 \rceil$. We will prove that with probability at least $1/2$, $\mathcal{Q}$ satisfies the two desired properties as stated in Lemma A.1. This implies the existence of such set $\mathcal{P}$.

It is clear that $\|q_i\|_2 = 1$ for all $i \in [n]$, since each $q_i$ is drawn from the unit sphere. We now prove that for any $i, j \in [n]$ with $i \neq j$, with probability at least $1 - \frac{1}{n^2}$, we have $|\langle q_i, q_j \rangle| \leq \varepsilon$. Notice that this is sufficient to prove the lemma, since by a union bound over all the $\binom{n}{2} = n(n-1)/2$ possible pairs of $(i, j)$, this implies that $\mathcal{Q}$ satisfies the two desired properties with probability at least $1/2$.

Now, we prove that for two independent random unit vectors $u$ and $v$ in $\mathbb{R}^d$ with $d = \lceil 8 \ln n/\varepsilon^2 \rceil$, with probability at least $1 - \frac{1}{n^2}$, $|\langle u, v \rangle| \leq \varepsilon$. By rotational invariance, we assume that $v$ is a standard basis vector. I.e., we assume $v_1 = 1$ and $v_i = 0$ for all $1 < i \leq d$. Notice that now $\langle u, v \rangle$ is the magnitude of the first coordinate of $u$. We finish the proof by invoking Lemma B.1 and taking $\beta = 8 \ln n > 6$. □

## B.3 Proof of Lemma A.2

*Proof of Lemma A.2.* Consider a $\sqrt{\epsilon}$-packing $\mathcal{N}$ with size $\Omega(1/\epsilon^{d/2})$ on the $d$-dimensional unit sphere $\mathbb{S}^{d-1}$ (for the existence of such a packing, see, e.g., Lorentz (1966)). Let $o$ be the origin. For two points $x, x' \in \mathbb{R}^d$, we denote $|xx'| := \|x - x'\|_2$ the length of the line segment between $x, x'$. Note that every two points $x, x' \in \mathcal{N}$ satisfy $|xx'| \geq \sqrt{\epsilon}$.

To prove the lemma, it suffices to show that $\mathcal{N}$ satisfies the property *equation* 1. Consider a point $x \in \mathcal{N}$, let $A$ be a hyperplane that is perpendicular to $x$ (notice that $x$ is a also a vector) and separates $x$ and every other points in $\mathcal{N}$. We let the distance between $x$ and $A$ be the largest possible, i.e., $A$ contains a point in $\mathcal{N}\backslash\{x\}$. Since $x$ is on the unit sphere and $\mathcal{N}$ is a $\sqrt{\epsilon}$-packing, we have that $x$ is at least $\sqrt{\epsilon}$ away from every point on the spherical cap not containing $x$, defined by the cutting plane $A$. More formally, let $b$ be the intersection point of the line segment $ox$ and $A$. Then

$$\forall y \in \left\{y' \in \mathbb{S}^{d-s} : \langle b, y' \rangle \leq \|b\|_2^2\right\} : \quad \|x - y\|_2 \geq \sqrt{\epsilon}.$$

Indeed, by symmetry, $\forall y \in \{y' \in \mathbb{S}^{d-1} : \langle b, y' \rangle \leq \|b\|_2^2\}$,

$$\|x - y\|_2 \geq \|x - z\|_2 \geq \sqrt{\epsilon}.$$

where $z \in \mathcal{N} \cap A$. Notice that the distance between $x$ and the convex hull of $\mathcal{N}\backslash\{x\}$ is lower bounded by the distance between $x$ and $A$, which is given by $|bx|$. Consider the triangles defined by $x, z, o, b$. We have $bz \perp ox$ (note that $bz$ lies inside $A$). By Pythagorean theorem, we have

$$|bz|^2 + |bx|^2 = |xz|^2;$$
$$|bx| + |bo| = |xo| = 1;$$
$$|bz|^2 + |bo|^2 = |oz|^2 = 1.$$

Solve the above three equations for $|bx|$, we have

$$|bx| = |xz|^2/2 \geq \epsilon/2$$

as desired. □

## C    EXACT LINEAR $Q^*$ + GAP IN GENERATIVE MODEL

In this section we present and prove the following theorem.

**Theorem C.1.** *Under Assumption 3.1, Assumption 4.2 and* **Generative Model** *query model, the agent can find the optimal $\pi^*$ with* $\text{poly}\left(d, H, \frac{1}{\rho}, \log\left(\frac{1}{\delta}\right)\right)$ *queries with probability $1 - \delta$ for a given failure probability $\delta > 0$,*

*Proof of Theorem C.1.* We first describe the algorithm. For each level, the agent first construct a barycentric spanner $\Lambda_h \triangleq \{\phi(s_h^1, a_h^1), \ldots \phi(s_h^d, a_h^d)\} \subset \Phi_h \triangleq \{\phi(s, a)\}_{s \in \mathcal{S}_h, a \in \mathcal{A}}$ (Awerbuch & Kleinberg, 2008). We have the property that any $\phi(s, a)$ with $s_h \in \mathcal{S}_h, a \in \mathcal{A}$, we have $c_{s,a}^1, \ldots, c_{s,a}^d \in [-1, 1]$ such that $\phi(s, a) = \sum_{i=1}^d c_{s,a}^i \phi(s_h^i, a_h^i)$.

The algorithm learns the optimal policy from $h = H - 1, \ldots, 0$. At any level $h$, we assume the agent has learned the optimal policy $\pi_{h'}^*$ at level $h' = h + 1, \ldots, H - 1$.

Now we present a procedure to show how to learn the optimal policy at level $h$. At level $h$, the agent queries every vector $\phi(s_h^i, a_h^i)$ in $\Lambda_h$ for $\text{poly}(d, \frac{1}{\rho}, \log\left(\frac{H}{\delta}\right))$ times and uses $\pi_{h+1}^*, \ldots, \pi_H^*$ as the roll-out to get the on-the-go reward. Note by the definition of $\pi^*$ and $Q^*$, the on-the-go reward is an unbiased sample of $Q^*(s_h^i, a_h^i)$. We denote $\widehat{Q}(s_h^i, a_h^i)$ the average of these on-the-go rewards. By Hoeffding inequality, it is easy to show with probability $1 - \frac{\delta}{H}$, for all $i = 1, \ldots, d$, $\left|\widehat{Q}(s_h^i, a_h^i) - Q^*(s_h^i, a_h^i)\right| \leq \text{poly}\left(\frac{1}{d}, \rho\right)$. Now we define our estimated $Q^*$ at level $h$ as follow: for any $(s, a) \in \mathcal{S}_h \times \mathcal{A}$, $\widehat{Q}(s, a) = \sum_{i=1}^d c_{s,a}^i \widehat{Q}(s_h^i, a_h^i)$. By the boundedness property of $c_{s,a}$, we know for any $(s, a) \in \mathcal{S}_h \times \mathcal{A}$, $\widehat{Q}(s, a) - Q^*(s, a) < \frac{\rho}{2}$. Note this implies the policy induced by $\widehat{Q}$ is the same as $\pi^*$. Therefore by induction we finish the proof.

$\square$

## D    LINEAR $Q^\pi$ FOR ALL $\pi$ IN GENERATIVE MODEL

In this section we present and prove the following theorem.

**Theorem D.1.** *Under Assumption 4.2 with $\delta = 0$, in the* **Generative Model** *query model, there is an algorithm that finds an $\epsilon$-optimal policy $\hat{\pi}$ using* $\text{poly}\left(d, H, \frac{1}{\epsilon}\right)$ *trajectories with probability 0.99.*

*Proof of Theorem D.1.* The algorithm is the same as the one in Theorem C.1 We only need to change the analysis. Suppose we are learning at level $h$ and we have learned policies $\pi_{h+1}, \ldots, \pi_{H-1}$ for level $h + 1, h + 2, \ldots, H - 1$, respectively. Because we use the roll-out policy $\pi_{h+1} \circ \cdots \circ \pi_{H-1}$, by Assumption 4.2 and the property of barycentric spanner, using the same argument in the proof of Theorem C.1, we know with probability $1 - 0.01/H$, we can learn a policy $\pi_h$ with $\text{poly}\left(d, H, \frac{1}{\epsilon}\right)$ samples such that for any $s \in \mathcal{S}_h$, we know $\pi_h$ is only sub-optimal by $\frac{\epsilon}{H}$ from the $\tilde{\pi}_h$ where $\tilde{\pi}_h$ is the optimal policy at level $h$ such that $\pi_{h+1} \circ \cdots \circ \pi_{H-1}$ is the fixed roll-out policy.

Now we can bound the sub-optimality of $\hat{\pi} \triangleq \pi_0 \circ \cdots \circ \pi_{H-1}$:

$$V^{\pi_0 \circ \pi_1 \circ \cdots \circ \pi_{H-1}}(s_1) - V^{\pi_0^* \circ \pi_1^* \circ \cdots \circ \pi_{H-1}^*}(s_1)$$

$$= V^{\pi_0 \circ \pi_1 \circ \cdots \circ \pi_{H-1}}(s_1) - V^{\tilde{\pi}_0 \circ \pi_1 \circ \cdots \circ \pi_{H-1}}(s_1)$$

$$+ V^{\tilde{\pi}_0 \circ \pi_1 \circ \cdots \circ \pi_{H-1}}(s_1) - V^{\pi_0^* \circ \pi_1 \circ \cdots \circ \pi_{H-1}}(s_1)$$

$$+ V^{\pi_0^* \circ \pi_1 \circ \cdots \circ \pi_{H-1}}(s_1) - V^{\pi_0^* \circ \pi_1^* \circ \cdots \circ \pi_{H-1}^*}(s_1).$$

The first term is at least $-\frac{\epsilon}{H}$ by our estimation bound, The second term is positive by definition of $\tilde{\pi}_0$. We can just recursively apply this argument to obtain

$$V^{\pi_0 \circ \pi_1 \circ \cdots \circ \pi_{H-1}}(s_1) - V^{\pi_0^* \circ \pi_1^* \circ \cdots \circ \pi_{H-1}^*}(s_1)$$

$$\geq V^{\pi_0^* \circ \pi_1 \circ \cdots \circ \pi_{H-1}}(s_1) - V^{\pi_0^* \circ \pi_1^* \circ \cdots \circ \pi_{H-1}^*}(s_1) - \frac{\epsilon}{H}.$$

$$\geq V^{\pi_0^* \circ \pi_1^* \circ \cdots \circ \pi_{H-1}}(s_1) - V^{\pi_0^* \circ \pi_1^* \circ \cdots \circ \pi_{H-1}^*}(s_1) - \frac{2\epsilon}{H}.$$

$$\geq \ldots$$
$$\geq - \epsilon.$$

$\square$

