# OpenReview forum: "Is a Good Representation Sufficient for Sample Efficient Reinforcement Learning?"
_ICLR.cc/2020/Conference — Accept (Spotlight)_

### Official Review · AnonReviewer1 · 2019-10-18
**Official Blind Review #1**

**Rating:** 6

**Review:**

This paper's contribution is a sample complexity lower bound for linear value-based learning and policy-based learning methods. The bound being exponential in the planning horizon is bad news, and has some implications with respect to further analysing sample complexity in RL.

The gist of this paper is that one can craft a hard MDP which requires visiting every state at least once, and that since this MDP's state space is exponential in the MDP's horizon, then there exists a set of MDPs which require an exponential (in the horizon) number of trajectories to be solved. As a consequence, further analysis of sample complexity in RL may need some much stronger assumptions.

The writing of the paper is good, I was able to understand everything (I think). As far as I can tell, this is novel work. Unfortunately I am currently unable to see why this contribution is valuable. I have set my score to weak reject but I am very open to having my mind changed, as I feel I may have missed some critical element.

I have two criticisms:
A- I don't understand why this bound is significantly different than previous bounds.
B- I don't understand why this is bad news for representation learning, nor how this failure mode of linear features translates to the "deep" case.

In the same spirit, I find rather odd the way the paper is introduced. Discussions of representations usually involve some discussion of generalization, but that's not what this paper is about. Deep neural networks/representation learning are only useful if there is an opportunity for generalization.


With respect to A, I am either grossly misunderstanding past bounds and/or your bounds, or something is wrong with the way complexities are compared:
- In Wen & Van Roy, the "polynomial" sample complexity is in the number of states, it is related to |S|x|A|xH^2 (Theorem 3 of Wen & Van Roy)
- In this paper, Theorem 4.1 states that the sample complexity is exponential because it is of the form 2^H. One *critical* assumption for this bound is precisely that |S| >= 2^H. Thus the bound that you propose is still polynomial in |S|.
I am thus puzzled, how is this bound significantly different?


With respect to B, I don't see how this bound has much to do with good representations, or even representations at all.
In Lemma A.1, you essentially craft a set of features that, being mutually orthogonal, are in some sense "mutually linearly separable", making learning the mapping from those features to a value function "trivial" once data is obtained. This is barely different from saying that you assume there is a magical learner that learns in O(1) given the data, because in either case, you need to visit _every_ of the 2^H state in order to solve the MDP, because by construction of your problem, there is _no hope_ of generalization*. Since learning features or creating "good" features has everything to do with generalization (otherwise we'd just to tabular), I don't see how this bound is relevant to representations. (We already have Wolpert's no free lunch theorem to tell us that there are always some problems that ML just can't be general enough to solve efficiently. What is more interesting is understanding how we can efficiently learn where there _is_ structure to a problem.)
* There is no hope of generalization, unless something about the observation space (which is left undiscussed in the paper) contains *information* about the agent being to the unique path to the reward. In such a case, I can see a probabilistic argument being made where in the worst case the agent needs to visit all 2^H states, but in the average case, the agent may learn to ignore paths where it can generalize that there is no reward. This is not entirely unreasonable, think of e.g. AlphaGo, where very few states end in victory, where there is an exponential number of states in the horizon, yet learning is totally reasonable because of structure in observation. This is where I don't agree with a statement like: "Since the class of linear functions is a strict subset of many more complicated function classes, including neural networks in particular, our negative results imply lower bounds for these more complex function classes as well."


**Experience Assessment:**

I have read many papers in this area.

**Review Assessment: Checking Correctness Of Derivations And Theory:**

I assessed the sensibility of the derivations and theory.

**Review Assessment: Checking Correctness Of Experiments:**

N/A

**Review Assessment: Thoroughness In Paper Reading:**

I read the paper thoroughly.

---

> ### Author Response · Authors · 2019-11-09
> **Response**
>
> We thank the reviewer for agreeing our paper is novel. First, we would like to clarify that this paper focuses on reinforcement learning with function approximation, where the goal is to understand when the sample complexity needed is much lower than the size of the state space. We would like to emphasize that there is a rich literature on conditions which permit sample efficient reinforcement learning, but there are (to our knowledge) no results on what is necessary.
>
> Please find our responses to your two criticisms below. The reviewer is indeed missing some of fundamental mathematical points; the reviewer does specifically mention that he/she might be mistaken on these points. We will try to clarity these points.
>
> To criticism A: “misunderstanding past bounds and/or your bounds”
> The reviewer is not interpreting the result of Wen & Van Roy correctly (cf.  https://papers.nips.cc/paper/4972-efficient-exploration-and-value-function-generalization-in-deterministic-systems.pdf ); the relevant case is the part “Polytopic prior constraint” which has O(H*d) sample complexity, where d is the dimension of features, and the size of the state space |S| could actually be as large as 2^H here. The reviewer is currently referring to the tabular case in Wen and Roy’s paper which has no function approximation. In our paper, Theorem 4.1 states that the sample complexity is exponential because it is of the form 2^H, and this shows the brittleness of the assumptions that the optimal Q-function can be *perfectly* predicted by the given representation in Wen and Van Roy’s paper. Thus, our bounds are exponentially different. Please let us know if this is not clear.
>
> Also, more generally, note that, same as our paper, Wen and Van Roy’s paper also focuses on RL with function approximation, where they are focusing on sufficient conditions for sample efficient RL.
>
>
> To criticism B:
> We believe the reviewer is not correct in the analogue of Wolpert’s “no free lunch theorem” here because our lower bound is not simply showing that learning in a binary tree is information-theoretically hard. Our lower bound is a stronger claim about learning with function approximation. Specifically, we are considering a function approximation setting where we assume that we have low dimensional features that well-approximates the value of every policy. In our RL lower bound, we show that even with this strong assumption, generalization is still not possible and requires 2^H samples.
>
> Let us try to clarify one potential source of confusion. We definitely agree that learning a binary tree of depth H is information-theoretically difficult with no further assumptions (as correctly pointed out by the reviewer). However, the key to our constructed lower bound is in fact a precise mathematical reduction showing that trying to learn with a good low dimensional representation is in fact no easier than learning a binary tree of depth H, where we know the latter is clearly information-theoretically difficult. In contrast, in supervised learning, we know that linear regression is doable with O(d) samples, and Wolpert’s “no free lunch theorem” is not relevant in the linear regression setting (when we assume that regression function well-approximates the target function, as we do in our setting).
>
> From a technical standpoint, the proofs require non-trivial constructions of features, which to our knowledge, are the first set of analyses of such kind. Therefore, the lower bound with function approximation is significantly different from existing bounds for tabular RL. Furthermore, for the value-based learning lower bound, we establish an intriguing connection between reinforcement learning with function approximation and approximate rank. For policy-based learning, we used lower bound on the size of eps-nets. We believe these technical ideas will be useful for other theoretical RL problems.

---

> > ### Comment · AnonReviewer1 · 2019-11-12
> > **Further questions**
> >
> > Thank you for your clarifications, they helped me understand a few things and I changed my score.
> >
> > wrt A, this makes a lot more sense than how I previously understood the bounds.
> >
> > wrt B, I think I now better understand what makes this proof interesting. I by no means want to diminish the effort that was put into the proofs, I don't believe they are trivial and did not intend in my review to give this impression.
> > That being said, I am still confused over the scope of these theorems.
> >
> > To the best of my knowledge, linear regression isn't always doable with O(d) samples. It _is_ with a few mild assumptions that work _most_ of the time, but there are cases where the curse of dimensionality takes over and 2^d samples are required. A great example is the one that is in the paper. If the data distribution is such that any two samples are always either nearly orthogonal or nearly the same, and if the optimal parameters are nearly orthogonal to all except one mode of the data, then I am fairly confident that with standard noise assumptions linear regression in this case will take O(2^d) samples.
> > Now, what happens if we jiggle and widen the data distribution to make it less orthogonal? I'd argue that this is a more sensible and realistic setting.
> > In the linear regression case, because we assume that the true X->Y mapping is linear, then we should be back to O(d) samples.
> > In the value prediction case, I'm not sure what happens. As far as I understand it, this would be beyond the scope of the assumptions in this paper, as the resulting MDP's features would no longer satisfy Lemma A.1 Am I understanding this correctly?
> >
> > If I may go back to this central question, "Is a good representation sufficient for sample-efficient reinforcement learning?" In the DNN literature, what are considered good representations aren't simply orthogonally representing factors of variations such that any two latent representations are orthogonal. Such representations would be considered quite bad, since what makes the strength of good reprentation is that interpolations in the latent space generally "make sense" (think about all those VAE/GAN papers showing linear interpolations in the latent space producing consistent images).
> > Qualifying the very orthogonal representations that are built in this paper as "good" is somewhat of a stretch from this point of view, since one couldn't use those representations to answer any other question or to make any [linear] interpolations. Is this correct?
> >
> > I realise I am very much out of my comfort zone, and at this point I am asking these questions more out of curiosity. Hopefully this can help you make your paper more accessible to neophytes.

---

> > > ### Author Response · Authors · 2019-11-15
> > > **Response to further questions**
> > >
> > > We are glad that our clarification helps your understanding. Please find our responses to your questions below.
> > >
> > > Linear regression is actually doable with $O(d)$ samples in the setting studied in this paper. As mentioned on Page 7, the features constructed in Lemma A.1 are in fact random unit vectors. Standard results for linear regression (see, e.g., Theorem 1 in https://arxiv.org/abs/1106.2363) implies that for such features, linear regression is in fact doable.
> > >
> > > To your other comments:
> > >
> > > “Now, what happens if we jiggle and widen the data distribution to make it less orthogonal……... as the resulting MDP's features would no longer satisfy Lemma A.1 Am I understanding this correctly?”
> > >
> > >
> > > Yes, the resulting MDP does not satisfy Lemma A.1. In this paper, our main message is that the assumption (the features can approximate the optimal policy or the value function well) is not sufficient for efficient RL. Studying assumptions on the features that permit efficient RL is an interesting problem to pursue.
> > >
> > >
> > > “If I may go back to this central question, …… since one couldn't use those representations to answer any other question or to make any [linear] interpolations. Is this correct?”
> > >
> > >
> > > This is not correct. We can still use the features for interpolation to do value function prediction. This is precisely what Assumption 4.1 and Assumption 4.2 provide.

---

### Official Review · AnonReviewer3 · 2019-10-23
**Official Blind Review #3**

**Rating:** 8

**Review:**


This paper presents theoretical lower bounds on sample complexities to learn good policies in reinforcement learning. The derived theorems show that there exists MDPs which require an exponential number of samples to learn a near-optimal policy even if a good-but-not-perfect representation is given to the agent for both value-based and policy-based learning. These results constitute the first lower bounds for RL with linear function approximation.

Representation learning is an important area of research and this paper advances our theoretical understanding in a notable way, helping to elucidate the limits of representation learning alone. The lower bounds derived in the paper would be of particular interest to the community as they can apply to a wide range of function approximators, including neural networks. Although this is not my area, the contributions are well-explained in the context of previous work and the theory was fairly easy to follow. The discussion also contained interesting points and summarized possible implications of the theoretical results. Overall, I think this paper presents a solid contribution and recommend acceptance.

Although the paper was clear in general, I would like to have certain points clarified:
1) Just to check, is it still possible that there exists certain representations that are not perfect but do lead to sample efficient learning? If I'm interpreting the results correctly, the theorems only posit the existence of a representation that is good in the sense of approximation error, but bad in terms of sample complexity, which does not necessarily preclude the possibility of other efficient representations.
2) More generally, why is it that there are few results for lower bounds when it seems like an obvious direction? Are there technical barriers to proving such results?
3) For value-based learning, the good representation has approximation error \Omega(\sqrt(H / d)). Could the authors explain why this assumption on the error is reasonable?
3) In assumption 4.4, the features are assumed to have an l2-norm of 1. This seems like a fairly restrictive assumption. How important is this assumption and can it be relaxed?
4) Also, in theorem 4.2, it is assumed that the dimension of the features, d = H. This would seem to allow the possibility of policy-based learning being sample efficient when the number of features is much smaller.

The paper is well-polished with few noticeable typos.
Minor comments:
- p.5 sec3.3: "knows the whole transition" -> "knows the whole transition function"
- p.8 sec5.1: "our lower bound on policy-based learning thus demonstrates" It may be worth reminding the reader that the bound applies to perfect representations -> "our lower bound on policy-based learning---which applies to perfect representations---thus demonstrates"

**Experience Assessment:**

I have read many papers in this area.

**Review Assessment: Checking Correctness Of Derivations And Theory:**

I assessed the sensibility of the derivations and theory.

**Review Assessment: Checking Correctness Of Experiments:**

N/A

**Review Assessment: Thoroughness In Paper Reading:**

I read the paper at least twice and used my best judgement in assessing the paper.

---

> ### Author Response · Authors · 2019-11-09
> **Response**
>
> We would like to thank the reviewer for the encouraging comments. Please find our response to your questions below.
> 1)	In this paper, we focus on good representation for policy-based learning and value-based learning, which are the two most widely used reinforcement learning paradigms in practice. With the additional assumption that the representation encodes information of the transition model, it has been shown that efficient reinforcement learning is indeed possible [1, 2]. However, it is unclear whether such an assumption holds in practice. Meanwhile, the lower bound in Theorem 4.1 actually implies that for the hard instance, *almost all* representations are hard. See the paragraph before ``Lower bound for policy-based learning’’ on Page 7.
> 2)	First, our lower bounds are different from those in the tabular setting since we need to take function approximation into consideration. Also, notice that when the planning horizon H = 1, policy-based/value-based RL learning are in fact equivalent to linear classification/linear regression, which do admit polynomial sample complexity upper bounds. Thus, our lower bounds critically rely on the fact that H is large. To utilize this, we need constructions and techniques different from those in supervised learning and bandit problems. As we have discussed in the paper, our lower bound is due to a distribution mismatch in the sense that the agent does not know which distribution to use for minimizing a (supervised) learning error.
> 3)	In many scenarios, the representation is the last layer of a neural network. Modern neural networks are often over-parameterized, which makes $d$ large. In this case, $\Omega(\sqrt{H / d})$ is actually very small. Thus, our lower bound holds even if the representation has small approximation error.
> 4)	In this paper, we focus on proving lower bounds. Thus, the more restrictive the assumptions are, the stronger the lower bound is. Using Assumption 4.4 as an example, since our lower bound holds if even the feature vectors are required to have unit l2 norms, for the more general case that different feature vectors could have different l2 norms, the lower bound still holds.
> 5)	This assumption is only for simplicity. We will modify the statement, and the lower bound will become $2^{\min\{d,H\}}$.
>
> Typos: We will fix them soon. Thanks for pointing out.
>
> [1] Lin F. Yang and Mengdi Wang. Sample-optimal parametric q-learning using linearly additive features. ICML 2019.
> [2] Chi Jin, Zhuoran Yang, Zhaoran Wang, and Michael I. Jordan. Provably efficient reinforcement learning with linear function approximation. arXiv 2019.

---

### Official Review · AnonReviewer2 · 2019-10-23
**Official Blind Review #2**

**Rating:** 8

**Review:**


The author question an important aspect which is very often taken for granted in the RL community, that a good representation could lead to data-efficient RL. They show negative results, providing pessimistic lower bounds for both value-based and policy-based learning.

I believe the paper is an important contribution, in particular, it has the following advantages:

 - Well written, clear, and nicely structured. It is self contained, with main results and convincing sketch proofs provided in the main body of the document, and extended technical proofs made available in the supplementary material.

 - Authors provide the relevant related work and elegantly show how their work connects to the existing literature.

- The notation is consistent throughout the document, with necessary assumptions clearly stated.

- The discussion highlights important findings, in particular the difference between value-based and policy-based learning. Additionally, it offers some hope for sample-efficient RL, by discussing the exponential separation between policy-based RL and imitation learning, reminding the community that sample-efficient RL can still be achieved by IL even if it can’t be achieved through good-but-not-perfect representation.

   Minor comment: - Phrasing of Assumption 4.3 seems to be off.

**Experience Assessment:**

I have read many papers in this area.

**Review Assessment: Checking Correctness Of Derivations And Theory:**

I did not assess the derivations or theory.

**Review Assessment: Checking Correctness Of Experiments:**

N/A

**Review Assessment: Thoroughness In Paper Reading:**

I read the paper at least twice and used my best judgement in assessing the paper.

---

> ### Author Response · Authors · 2019-11-09
> **Response**
>
> We would like to thank the reviewer for the encouraging comments.
> We will polish the paper soon.

---

### Author Response · Authors · 2019-11-12
**Summary of Revision**

Following reviewers' suggestions, we have updated the paper and uploaded a revision on Nov 11. Here we give a summary of the major changes.

1. We add more explanation for Assumption 4.3. Now we explicitly state that this assumption is saying that the optimal policy is linear.
2. On Page 6, we explain why $\sqrt{H/d}$ is small in practice.
3. Theorem 4.2 does not require $d = H$ now, and the lower bound becomes $\Omega(\min\{2^H, 2^d\})$. We have also revised the corresponding part of the proof.
4. We have fixed typos found by Reviewer #3.

---

### Decision · Program_Chairs · 2019-12-19

**Decision:**

Accept (Spotlight)

**Comment:**

The authors challenge the idea that good representation in RL lead are sufficient for learning good policies with an interesting negative result -- they show that there exist MDPs which require an exponential number of samples to learn a near-optimal policy even if a good-but-not-perfect representation is given to the agent for both value-based and policy-based learning.  Reviewers had some minor technical questions which were clarified sufficiently by the authors, leading to a consensus of the contribution and quality of this work.  Thus, I recommend this paper for acceptance.